# Understanding inequalities in COVID-19 outcomes following hospital admission for people with intellectual disability compared to the general population: a matched cohort study in the UK

R Asaad Baksh ![ORCID] ,[1,2] Sarah E Pape,[1,2,3] James Smith,[3] André Strydom[1,2,3]

¹Institute of Psychiatry, Psychology, and Neuroscience, King's College London, London, UK
²The LonDowns Consortium, London, UK
³South London and Maudsley NHS Foundation Trust, London, UK

**Correspondence to**
Dr R Asaad Baksh;
asaad.baksh@kcl.ac.uk

## ABSTRACT

**Objectives** This study explores the hospital journey of patients with intellectual disabilities (IDs) compared with the general population after admission for COVID-19 during the first wave of the pandemic (when demand on inpatient resources was high) to identify disparities in treatment and outcomes.

**Design** Matched cohort study; an ID cohort of 506 patients were matched based on age, sex and ethnicity with a control group using a 1:3 ratio to compare outcomes from the International Severe Acute Respiratory and emerging Infections Consortium WHO Clinical Characterisation Protocol UK.

**Setting** Admissions for COVID-19 from UK hospitals; data on symptoms, severity, access to interventions, complications, mortality and length of stay were extracted.

**Interventions** Non-invasive respiratory support, intubation, tracheostomy, ventilation and admission to intensive care units (ICU).

**Results** Subjective presenting symptoms such as loss of taste/smell were less frequently reported in ID patients, whereas indicators of more severe disease such as altered consciousness and seizures were more common. Controls had higher rates of cardiovascular risk factors, asthma, rheumatological disorder and smoking. ID patients were admitted with higher respiratory rates (median=22, range=10–48) and were more likely to require oxygen therapy (35.1% vs 28.9%). Despite this, ID patients were 37% (95% CI 13% to 57%) less likely to receive non-invasive respiratory support, 40% (95% CI 7% to 63%) less likely to receive intubation and 50% (95% CI 30% to 66%) less likely to be admitted to the ICU while in hospital. They had a 56% (95% CI 17% to 102%) increased risk of dying from COVID-19 after they were hospitalised and were dying 1.44 times faster (95% CI 1.13 to 1.84) compared with controls.

**Conclusions** There have been significant disparities in healthcare between people with ID and the general population during the COVID-19 pandemic, which may have contributed to excess mortality in this group.

## STRENGTHS AND LIMITATIONS OF THIS STUDY

⇒ This is the first in-depth analysis of the hospital journey of patients with intellectual disabilities compared with the general population after admission for COVID-19.

⇒ We had a large sample size of 506 patients with intellectual disabilities and 1518 well-matched controls.

⇒ Our dataset included data on comorbidities, vital signs, COVID-19 related admission signs and symptoms, complications due to COVID-19, information regarding interventions and outcome of hospitalisation.

⇒ Due to data being collected at the time of care there was some degree of missing or incomplete data.

## INTRODUCTION

Intellectual disability (ID) is a condition characterised by varying degrees of impairments in cognition, language, motor and social abilities depending on the severity of ID[1] and affects around 1% of the population globally.[2] Poorer health outcomes compared with the general population have been consistently reported for people with ID,[3] with an increased incidence of comorbidities including dysphagia and respiratory diseases, with respiratory disease identified as a leading cause of death.[4] These health comorbidities are associated with poor outcomes following infections and other acute conditions,[5 6] which may be exacerbated by barriers in accessing health and social care, associated with concerns about ongoing discrimination and bias.[7]

To date there have been over 64 million cases of COVID-19 reported worldwide and 1.4 million deaths.[8] Several risk factors for increased mortality have been identified and

reported,[9] including increasing age,[10] cardiovascular disease, chronic lung disease,[11] cancer,[12] chronic kidney disease[13] and obesity.[14] Evidence is now emerging that people with ID are disproportionally negatively impacted by COVID-19.[6 15 16] The number of deaths of people with ID in England was three times higher in 2020 when compared with the corresponding period 2 years before[17] and people with ID may be more seriously affected by COVID-19 at a younger age than the general population.[15 18] Those with Down syndrome may be at particular risk of a more severe disease course,[19–21] specifically those 40 years and older.[22] Recent research has also suggested that people with Down syndrome have an increased risk of COVID-19 hospitalisation and death.[23]

Given the existing health inequalities for people with ID, it is reasonable to further examine how people with COVID-19 and ID present to and progress through the acute hospital system and how this compares to the experiences of the general population. To date, only a few small-scale studies have examined the clinical presentation of COVID-19 in people with ID[15 16] and none have provided a comprehensive picture of their experiences once admitted to hospital for COVID-19. Specifically, there is little evidence relating to resources and treatment allocation for people with ID and how this compares to the general population.

Decisions around escalation of care, for example to intensive care units (ICUs), are complicated during a pandemic with added pressures related to rationing of resources. Such decisions have come under increasing scrutiny during the COVID-19.[24 25] In the UK the National Health Service offered guidance to hospital trusts related to resource allocation,[26] however, there is little research about the impact of these guidelines on vulnerable populations such as people with ID.

The aim of our study was to explore the hospital journey of patients with ID compared with the general population after they were admitted to hospital for COVID-19 during the first wave of the pandemic, when pressure on healthcare systems were most acute. We have chosen to focus on interventions that require triaging and resource allocation, for both clinical and supply reasons[26–28]: non-invasive ventilation (NIV), tracheal intubation and admission to ICU. Comparisons were made to the general population in the following areas: (1) pattern and severity of COVID-19 symptoms at time of hospital admission; (2) comorbidities; (3) admission to intensive care and use of intubation and/or ventilation treatments; (4) complications during hospital admission; (5) outcomes of admission including length of stay and mortality.

## METHOD
### Study design and setting
This study used data from the International Severe Acute Respiratory and emerging Infections Consortium (ISARIC) WHO Clinical Characterisation Protocol UK (CCP-UK). The ISARIC4C CCP-UK is an ongoing prospective cohort study in 260 hospitals across England, Scotland and Wales (National Institute for Health Research Clinical Research Network Central Portfolio Management System ID 14152).[9] The ISARIC4C CCP-UK protocol was activated on 17 January 2020 and information regarding the protocol, supplementary documents and details of the Independent Data and Material Access Committee (IDAMAC) are available online (https://isaric4c.net).

### Participants
The inclusion criteria for enrolment into the ISARIC4C CCP-UK cohort were patients of any age who were admitted to acute care hospitals with a proven or high likelihood of SARS-CoV-2 infection. Patients were admitted to hospital at the discretion of their clinical team and the study authors did not set criteria for inclusion. Patients who were already admitted to hospital for a separate clinical reason but had subsequently tested positive for COVID-19 during their stay were also included in the present study.[9]

Overall, in our sample were a total of 59 025 patients who were admitted between February 2020 and 9 July 2020 (downloaded on 24 July 2020). We identified 566 (0.96%) patients who had a diagnosis of ID and matched these patients to general population controls in the same dataset by age group, sex and ethnicity using a 1:3 ratio of ID patients to controls with no duplication of controls. Of the 566 ID patients, 506 had complete data on age group, sex and ethnicity and were matched to 1518 general population controls.

### Data collection
Data were collected using a paper case report form that was developed by ISARIC4C CCP-UK and the WHO for use in outbreak investigations and uploaded to a REDCap database (Research Electronic Data Capture, Vanderbilt University, USA, hosted by University of Oxford, UK). Consent from patients was not required to collect anonymised demographic and clinical data for research in England and Wales. For patients in Scotland, a waiver for consent was given by the Public Benefit and Privacy Panel.

### Variables and data sources
Our dataset consisted of patient demographic information, comorbidities, vital signs, COVID-19 related admission signs and symptoms, complications due to COVID-19, information regarding interventions and outcome of hospitalisation. Data on these variables of interest were collected from the case report form developed by ISARIC4C CCP-UK and the WHO.

### Patient and public involvement
The ISARIC4C CCP-UK study was an urgent public health research study in response to a Public Health Emergency of International Concern, therefore patients were not involved in the design, conduct or reporting of this rapid response research.

## Bias and missing data

Criteria for the research team to enrol patients was based on local COVID-19 test reports, therefore the efficiency of testing labs may have biased patient enrolment. Data collection may have been limited by staff resources at times of high COVID-19 infections. Due to the timing and nature of the study, there were missing or incomplete data, particularly as infection rates grew exponentially in the UK. Missing data were not imputed in the present study and consequently complete data were not available for all variables.

## Data access and linkage

The study authors did not have direct access to the database population used to create the study population. Access to the study population data was granted by the Independent Data and Material Access Committee (https:// isaric4c.net). We did not conduct any data linkage for the present study.

## Statistical analysis

Descriptive statistics were used to show patient information, comorbidities and COVID-19 admission information, medical complications, interventions and outcomes. Statistical testing was performed using Fisher's exact test for frequency data while Mann-Whitney U was used for respiratory rate on admission and linear regression for frailty scores adjusted for age group and sex.

We conducted logistic regression modelling to examine whether demographic variables (age group and sex), severity of COVID-19 illness on admission (respiratory rate, need for oxygen therapy and shortness of breath), the number of comorbidities patients had on admission, a diagnosis of Down syndrome or an ID diagnosis were associated with COVID-19 related interventions. Similar logistic regression models were used to examine factors associated with mortality between groups, and with medical complications due to COVID-19. In the mortality between groups model we adjusted for significant mortality-related comorbidities for COVID-19 that have been previously reported in the ISARIC4C CCP-UK dataset; these included chronic cardiac disease, chronic pulmonary disease, chronic kidney disease, liver disease, obesity, chronic neurological disorder, dementia and malignant neoplasm.[9] We reported risk ratios (RRs) with 95% CIs for the logistic regression models. Time-to-event analysis using Cox proportional hazards modelling was used to examine how soon after admission patients with ID were dying from COVID-19 compared with controls while adjusting for covariates (age group, sex, severity of COVID-19 on admission, number of comorbidities and Down syndrome diagnosis). We used death as the outcome and data were depicted with a Kaplan-Meier curve. Finally, potential differences in length of stay between ID patients and controls were explored using linear regression adjusting for the same covariates as the Cox proportional hazards model. To avoid violation of normality, clinical frailty scores and days in hospital was log-transformed and back transformed for reporting. All data analyses were done using R V.4.0.3.

# RESULTS

## Description of study population and comorbidities

The sample of 506 ID patients consisted predominantly of adults over the age of 40 with only 25% of patients being under 40. Moreover, ID patients were mostly male and white, had lower rates of chronic cardiac disease, hypertension, chronic pulmonary disease, asthma, malignant neoplasm, and rheumatological disorders, and were less likely to be smokers than the general population controls (table 1). On the other hand, higher rates of chronic neurological disorders (a broad category including cerebral palsy, multiple sclerosis, motor neuron disease, muscular dystrophy, myasthenia gravis, Parkinson's disease, stroke, severe learning difficulty) were reported in ID patients compared with controls, with a higher prevalence of dementia. The increased dementia rate is likely secondary to the association between Down syndrome and Alzheimer's disease included in the ID group.

## Signs, symptoms and severity of illness on admission in hospitalised patients with COVID-19 with and without an ID diagnosis

A number of significant differences were observed in the symptoms at initial presentation to hospital between ID and control groups (table 2). In particular, subjectively reported signs and symptoms such as loss of taste/smell, as well as those related to pain (headache, chest pain and muscle aches) were all reported less frequently in patients with ID. On the other hand, altered consciousness or confusion (29.9% vs 17.6%) and seizures (9.9% vs 2.2%) were more common in patients with ID. Compared with controls, ID patients were admitted with higher respiratory rates and were more likely to require oxygen therapy. In addition, adjusted for age group and sex, having a diagnosis of ID was significantly associated with higher clinical frailty scores.

## Medical complications among hospitalised COVID-19 patients with and without an ID diagnosis

In both the ID and general population groups the leading complications due to COVID-19 (online supplemental appendix table 1) were pulmonary complications including viral pneumonia, bacterial pneumonia and acute respiratory syndrome, as well as acute renal injury and/or acute renal failure, anaemia and cardiac complications. Overall, medical complications were comparable between patients with ID and controls, with the exception of seizures which were more prevalent in the ID group (5.1% of those with ID compared with 2.0% of the control group).

**Table 1** Characteristics of patients hospitalised for COVID-19 with and without an ID diagnosis

| | | Controls | | ID group | | P value of comparison |
|---|---|---|---|---|---|---|
| | | n | N | n | N | |
| | | 1518 | | 506 | | |
| Age group (%) | <20 | 117 (7.7) | | 39 (7.7) | | |
| | 20–29 | 114 (7.5) | | 38 (7.5) | | |
| | 30–39 | 150 (9.9) | | 50 (9.9) | | |
| | 40–49 | 159 (10.5) | | 53 (10.5) | | |
| | 50–59 | 336 (22.1) | | 112 (22.1) | | |
| | 60–69 | 324 (21.3) | | 108 (21.3) | | |
| | 70–79 | 207 (13.6) | | 69 (13.6) | | |
| | 80+ | 111 (7.3) | | 37 (7.3) | | |
| Sex (%) | Female | 660 (43.5) | | 220 (43.5) | | |
| | Male | 858 (56.5) | | 286 (56.5) | | |
| Ethnicity (%) | Aboriginal/First Nations | 3 (0.2) | | 1 (0.2) | | |
| | Black | 36 (2.4) | | 12 (2.4) | | |
| | East Asian | 3 (0.2) | | 1 (0.2) | | |
| | Other | 96 (6.3) | | 32 (6.3) | | |
| | South Asian | 57 (3.8) | | 19 (3.8) | | |
| | West Asian | 9 (0.6) | | 3 (0.6) | | |
| | White | 1314 (86.6) | | 438 (86.6) | | |
| Chronic cardiac disease | | 309 (21.5) | 1439 | 81 (16.9) | 479 | **0.036** |
| Hypertension (physician diagnosed) | | 252 (31.9) | 791 | 56 (18.7) | 300 | **<0.001** |
| Chronic pulmonary disease (not asthma) | | 191 (13.4) | 1430 | 44 (9.2) | 478 | **0.016** |
| Asthma (physician diagnosed) | | 270 (18.8) | 1435 | 68 (14.1) | 481 | **0.022** |
| Chronic kidney disease | | 140 (9.8) | 1433 | 53 (11.0) | 481 | 0.432 |
| Mild, moderate or severe liver disease* | | 54 (3.8) | 1429 | 15 (3.1) | 480 | 0.574 |
| Diabetes† | | 266 (18.9) | 1407 | 85 (18.2) | 467 | 0.784 |
| Chronic neurological disorder | | 156 (10.9) | 1432 | 177 (36.6) | 483 | **<0.001** |
| Malignant neoplasm | | 100 (7.0) | 1426 | 20 (4.2) | 476 | **0.029** |
| Chronic haematological disease | | 39 (2.7) | 1427 | 13 (2.7) | 476 | 1.000 |
| Obesity (as defined by clinical staff) | | 207 (15.7) | 1317 | 69 (16.0) | 431 | 0.879 |
| Rheumatological disorder | | 99 (6.9) | 1426 | 20 (4.2) | 473 | **0.037** |
| Dementia | | 85 (5.9) | 1437 | 47 (9.9) | 473 | **0.005** |

**Table 1** Continued

| | Controls | | ID group | | P value of comparison |
|---|---|---|---|---|---|
| | n | N | n | N | |
| | 1518 | | 506 | | |
| Malnutrition | 30 (2.2) | 1378 | 12 (2.6) | 459 | 0.590 |
| Smoking | | | | | <0.001 |
| Former smoker | 279 (26.4) | 1055 | 43 (13.7) | 313 | |
| Never smoked | 676 (64.1) | | 247 (78.9) | | |
| Yes | 100 (9.5) | | 23 (7.3) | | |

The sample of 506 patients with an intellectual disability diagnosis from the UK ISARIC-4C matched to 1518 controls without an intellectual disability diagnosis based on age group, sex and ethnicity.
Significant differences between the ID group and controls are highlighted in bold.
*Mild, moderate and severe liver disease were combined into one category.
†The variables diabetes and type, diabetes with complications, and diabetes without complications were combined into one category. The number of patients in the ID group with the comorbidities listed above on admission to hospital were compared with controls using Fisher's exact test.
ID, intellectual disabilities.

## Factors associated with COVID-19 related interventions

An increased likelihood of admission to ICU, tracheal intubation and non-invasive respiratory support were all associated with higher respiratory rate, shortness of breath and the requirement of oxygen therapy on admission, suggesting that the severity of illness on admission is important for prognosis and the need for COVID-19 related interventions. Significantly fewer ID patients were admitted to ICU, underwent intubation or received non-invasive respiratory support compared with controls (table 3). Adjusted for age group, sex, severity of illness on admission, number of comorbidities and Down syndrome diagnosis, patients with ID were 37% less likely to receive non-invasive respiratory support, 40% less likely to receive intubation and 50% less likely to be admitted to the ICU while in hospital (figure 1).

## Mortality rates and factors associated with mortality among patients with COVID-19 with and without an ID diagnosis

People with ID had a 56% increased risk of dying from COVID-19 after they were hospitalised compared with controls, with a mortality rate of 29.2% for the ID group compared with 18.8% for controls (online supplemental appendix figure 1). Adjusted for age group, sex, known mortality-related comorbidities, severity of illness on admission, interventions and Down syndrome diagnosis, the association between mortality and an ID diagnosis remained significant (online supplemental appendix table 2).

Examining the factors associated with mortality in the ID group only we found that age (50 years and older), requiring oxygen therapy and higher respiratory rates at admission were all significantly associated with increased risk of dying from COVID-19. None of the known mortality-related comorbidities were significantly associated with mortality in patients with ID in our sample (online supplemental appendix table 3).

### Associations between medical complications and mortality

Viral pneumonia was significantly associated with mortality in the ID group. This complication increased ID patients' risk of dying by 174%. Acute respiratory syndrome was also strongly associated with mortality and increased ID patients' risk of dying by 107% (online supplemental appendix table 4).

In comparison, while still significantly associated with mortality in controls, viral pneumonia was associated with a 56% increase in risk of dying and acute respiratory syndrome increased risk of dying by 91%. On the other hand, cardiac arrest was associated with a 438% increase risk of dying in controls, gastrointestinal haemorrhage increased the risk of dying by 178%, acute renal injury by 99% and other cardiac complications by 82% (online supplemental appendix table 5).

Table 2  Admission signs, symptoms and severity of illness on admission related to COVID-19 in hospitalised patients with and without an ID diagnosis

| | Controls | | ID group | | P value of comparison |
|---|---|---|---|---|---|
| | n (%) | N | n (%) | N | |
| Cough | 972 (67.6) | 1438 | 309 (64.6) | 478 | 0.239 |
| Cough with sputum production* | 285 (22.7) | 1254 | 58 (14.6) | 397 | **<0.001** |
| Cough with bloody sputum | 41 (3.3) | 1240 | 9 (2.3) | 393 | 0.401 |
| Fever | 1004 (69.6) | 1442 | 335 (69.8) | 480 | 1.000 |
| Sore throat | 123 (10.4) | 1186 | 29 (8.0) | 364 | 0.191 |
| Runny nose* | 49 (4.2) | 1168 | 6 (1.7) | 357 | **0.023** |
| Wheezing | 94 (7.7) | 1228 | 41 (10.1) | 407 | 0.145 |
| Ear pain | 7 (0.6) | 1150 | 3 (0.8) | 364 | 0.711 |
| Chest pain* | 225 (17.8) | 1267 | 35 (8.7) | 404 | **<0.001** |
| Muscle aches* | 275 (23.1) | 1192 | 30 (8.4) | 357 | **<0.001** |
| Joint pain | 70 (6.1) | 1147 | 18 (5.1) | 356 | 0.520 |
| Fatigue | 511 (40.7) | 1254 | 145 (37.5) | 387 | 0.260 |
| Shortness of breath* | 953 (67.3) | 1416 | 274 (59.8) | 458 | **0.004** |
| Disturbance or loss of taste* | 51 (8.8) | 578 | 3 (1.4) | 207 | **<0.001** |
| Disturbance or loss of smell* | 36 (6.1) | 588 | 1 (0.5) | 212 | **<0.001** |
| Headache* | 177 (14.9) | 1184 | 20 (5.5) | 362 | **<0.001** |
| Altered consciousness or confusion* | 233 (17.6) | 1326 | 124 (29.9) | 415 | **<0.001** |
| Seizures | 28 (2.2) | 1291 | 41 (9.9) | 415 | **<0.001** |
| Abdominal pain | 187 (14.6) | 1280 | 53 (13.2) | 403 | 0.514 |
| Vomiting and/or nausea* | 323 (24.3) | 1329 | 67 (15.7) | 426 | **<0.001** |
| Diarrhoea* | 279 (21.0) | 1327 | 58 (13.4) | 432 | **<0.001** |
| Conjunctivitis | 11 (0.9) | 1205 | 4 (1.0) | 384 | 0.767 |
| Lymphadenopathy | 10 (0.8) | 1206 | 0 (0.0) | 390 | 0.131 |
| Skin rash | 33 (2.7) | 1228 | 8 (2.0) | 396 | 0.581 |
| Skin ulcers | 19 (1.5) | 1231 | 6 (1.5) | 401 | 1.000 |
| Haemorrhage | 19 (1.5) | 1261 | 4 (1.0) | 416 | 0.626 |
| Requirement of oxygen therapy on admission* | 406 (28.9) | 1407 | 170 (35.1) | 484 | **0.011** |
| Median respiratory rate (breaths per minute) on admission (IQR)** | 21 (10–50) | 1404 | 22 (10–48) | 464 | **0.009** |
| Mean clinical frailty score (SD) | 3.55 (2.17) | 437 | 5.14 (1.89) | 175 | **<0.0001** |

The number of patients in the ID group presenting with COVID-19 related symptoms on admission to hospital, compared with controls using Fisher's exact test.
*Significant difference between the ID group and controls.
Significant differences between the ID group and controls are highlighted in bold.
**We excluded respiratory rate values that were below 10 or higher than 50 breaths per minute as such data were considered outliers.
ID, intellectual disabilities.

## Survival analysis of patients with COVID-19 with and without an ID diagnosis

After 5 days in hospital, 16.6% of ID patients had died compared with only 6.5% of controls. This trend continued so that at 20 days 39.3% of ID patients had died compared with 32.7% of controls (online supplemental appendix table 6). Figure 2 shows the Kaplan-Meier estimates of survival probability for our ID group and controls. Adjusting for age group, sex, Down syndrome diagnosis, number of comorbidities and severity of COVID-19 on admission, the HR for COVID-19 related mortality in patients with ID compared with controls was 1.44 (95% CI 1.13 to 1.84, p=0.003). Therefore, patients with ID were dying 1.44 times faster than controls at any particular point in time after they were admitted to hospital for COVID-19, even after adjusting for covariates.

### Factors associated with length of time in hospital for patients with COVID-19 with and without an ID diagnosis

A significant association between a diagnosis of ID and length of time in hospital was found, with ID patients spending longer periods in hospital after they

**Table 3** COVID-19 related interventions for hospitalised patients with and without an intellectual disability diagnosis

| | Controls | | ID group | | |
|---|---|---|---|---|---|
| | n | N | n | N | P value of comparison |
| Non-invasive respiratory support | 243 (16.9) | 1436 | 60 (12.3) | 487 | **0.017** |
| Tracheal intubation | 167 (11.2) | 1496 | 36 (7.2) | 503 | **0.010** |
| Tracheostomy | 16 (2.5) | 637 | 2 (1.1) | 178 | 0.390 |
| Any time in intensive care unit | 304 (20.3) | 1500 | 59 (11.7) | 505 | **<0.001** |

Significant differences between the ID group and controls are highlighted in bold.
ID, intellectual disabilities.

were admitted for COVID-19 (table 4). The controls spent a mean of 10.98 days in hospital (SD=14.45, median=6.5 days) while the ID group spent 14.55 days on average (SD=13.29, median=11 days; online supplemental appendix figure 2). Other factors significantly associated with longer stays in hospital in both groups were being older than 20 years, more comorbidities and greater severity of illness on admission. An accessible summary of these results is presented in the (Infographic supplemental file).

## DISCUSSION

This is the first in-depth exploration of treatment and interventions offered to patients with ID who were admitted to hospital for COVID-19. We found that the hospital journey for people with ID and COVID-19 is substantially different to the general population in a number of fundamental areas: recognition and assessment of COVID-19 symptoms; symptoms and severity of illness on admission; access to interventions and ICUs; mortality rates, survival trajectories and duration of hospital stay.

### Recognition and assessment of COVID-19 symptoms

The most prevalent symptoms recorded at admission in both the ID and control group were cough, fever and shortness of breath, in keeping with previous reports.[29] However, patients with ID were significantly less likely to present with subjective symptoms including pain, loss of taste or smell and 'shortness of breath', despite having higher respiratory rates at admission. People with ID were more likely to present with altered consciousness, confusion and seizures which could indicate a more severe presentation on admission. Patients with ID also presented with other indicators of more severe illness at the point of admission, including greater requirement for supplemental oxygen therapy and increased average respiratory rates compared with controls. This could represent late presentation to hospital by people with ID. There are several potential explanations for late presentation of patients with ID: poor symptom recognition by caregivers and patients themselves, communication difficulties, and exclusion from digital information and public health campaigns which could reduce awareness about early warning signs and symptoms. Other issues

which may have contributed to later referral to hospital include a reluctance from family members to hospitalise their relative or disability discrimination resulting in people with ID not being able to access medical services.

### Course of illness in hospitalised patients with ID and access to interventions and ICUs

Once admitted, patients with ID and COVID-19 had a more aggressive course of disease, with higher rates of death in the early stages of hospitalisation as well as longer hospital stays. Rates of complications and most comorbidities were comparable between the groups, however patients with ID were given higher scores on the clinical frailty scale, potentially reflecting misinterpretation of the degree of frailty in the context of long-term but stable cognitive impairment. This has implications for treatment decisions around resource allocation when availability may be limited.

Despite having more severe symptoms on admission and similar rates of complications, patients with ID were less likely to be treated with NIV, tracheal intubation, or be admitted to an ICU setting. This disparity in access to appropriate treatment has been highlighted in investigations of other conditions,[30] with issues surrounding decision-making capacity, ceilings of care, inappropriate use of clinical frailty scales, and discrimination or biases potentially contributing to inequalities in care.[31] Other contributing factors may be related to tolerability of interventions (particularly non-invasive ventilation) for people with ID, perceived treatment difficulties that may influence decision making, and inappropriate use of Do Not Resuscitate orders.[32]

### Complications of COVID-19 infection, mortality rates, and length of stay

Having a diagnosis of ID was associated with a 56% increase in mortality risk, which was not associated with seizures or dementia, despite these conditions being more common in ID patients compared with the general population, particularly those with Down syndrome.[33] The increased mortality also does not appear to be related to other suggested COVID-19 comorbidities for adverse outcome,[9 11 13] although as in the general population, older age and severity of illness on admission did show significant associations with mortality in ID. As well as an

A

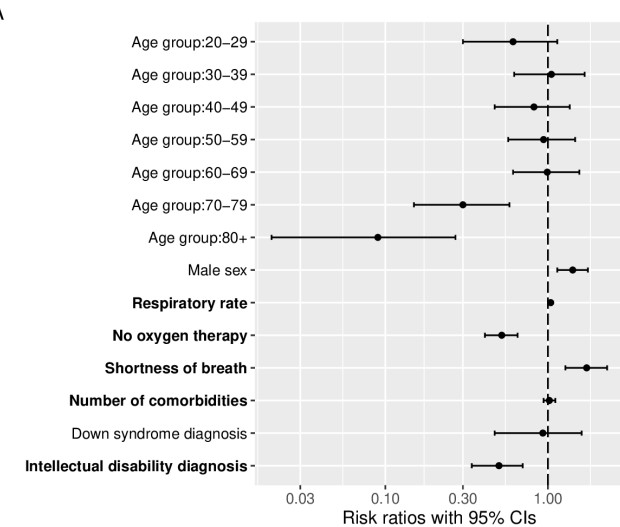

| | Risk ratio | Lower CI | Upper CI | p value |
|---|---|---|---|---|
| Age group:20–29 | 1.13 | 0.48 | 2.47 | 0.771 |
| Age group:30–39 | 1.31 | 0.63 | 2.62 | 0.474 |
| Age group:40–49 | 1.33 | 0.65 | 2.63 | 0.435 |
| Age group:50–59 | 1.73 | 0.93 | 3.18 | 0.095 |
| Age group:60–69 | 1.79 | 0.96 | 3.27 | 0.079 |
| Age group:70–79 | 0.83 | 0.39 | 1.77 | 0.630 |
| Age group:80+ | 0.50 | 0.19 | 1.26 | 0.147 |
| Male sex | 1.27 | 1.00 | 1.59 | 0.053 |
| Respiratory rate | 1.04 | 1.02 | 1.06 | <0.0001 |
| No oxygen therapy | 0.56 | 0.43 | 0.70 | <0.0001 |
| Shortness of breath | 2.10 | 1.50 | 2.93 | <0.0001 |
| Number of comorbidities | 1.14 | 1.05 | 1.24 | 0.002 |
| Down syndrome diagnosis | 1.38 | 0.76 | 2.22 | 0.261 |
| Intellectual disability diagnosis | 0.63 | 0.43 | 0.87 | 0.007 |

B

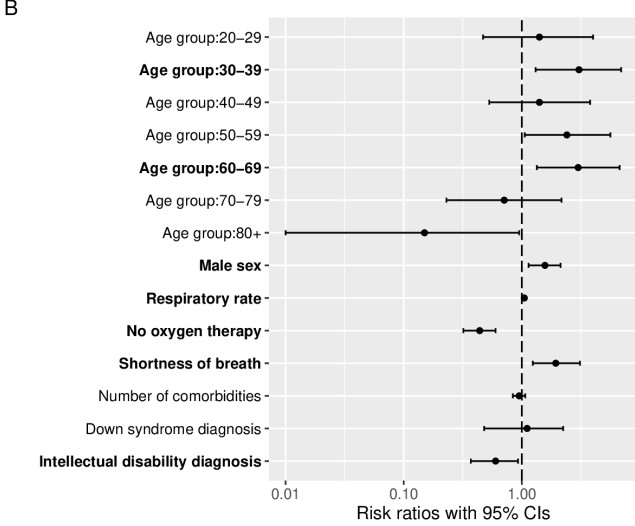

| | Risk ratio | Lower CI | Upper CI | p value |
|---|---|---|---|---|
| Age group:20–29 | 1.41 | 0.47 | 4.01 | 0.536 |
| Age group:30–39 | 3.05 | 1.31 | 6.93 | 0.015 |
| Age group:40–49 | 1.41 | 0.53 | 3.79 | 0.499 |
| Age group:50–59 | 2.41 | 1.06 | 5.63 | 0.051 |
| Age group:60–69 | 3.00 | 1.34 | 6.73 | 0.014 |
| Age group:70–79 | 0.71 | 0.23 | 2.17 | 0.538 |
| Age group:80+ | 0.15 | 0.01 | 0.95 | 0.086 |
| Male sex | 1.57 | 1.14 | 2.13 | 0.006 |
| Respiratory rate | 1.05 | 1.03 | 1.07 | <0.001 |
| No oxygen therapy | 0.44 | 0.32 | 0.60 | <0.0001 |
| Shortness of breath | 1.94 | 1.24 | 3.11 | 0.006 |
| Number of comorbidities | 0.95 | 0.84 | 1.07 | 0.396 |
| Down syndrome diagnosis | 1.11 | 0.48 | 2.24 | 0.795 |
| Intellectual disability diagnosis | 0.60 | 0.37 | 0.93 | 0.028 |

C

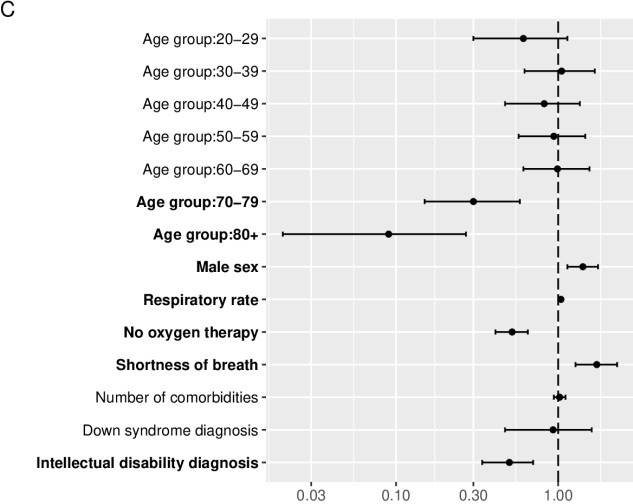

| | Risk ratio | Lower CI | Upper CI | p value |
|---|---|---|---|---|
| Age group:20–29 | 0.61 | 0.30 | 1.14 | 0.131 |
| Age group:30–39 | 1.05 | 0.62 | 1.68 | 0.852 |
| Age group:40–49 | 0.82 | 0.47 | 1.36 | 0.454 |
| Age group:50–59 | 0.94 | 0.57 | 1.47 | 0.786 |
| Age group:60–69 | 0.99 | 0.61 | 1.56 | 0.982 |
| Age group:70–79 | 0.30 | 0.15 | 0.58 | 0.0002 |
| Age group:80+ | 0.09 | 0.02 | 0.27 | <0.0001 |
| Male sex | 1.42 | 1.14 | 1.76 | 0.002 |
| Respiratory rate | 1.04 | 1.03 | 1.06 | <0.0001 |
| No oxygen therapy | 0.52 | 0.41 | 0.65 | <0.0001 |
| Shortness of breath | 1.73 | 1.28 | 2.31 | 0.0005 |
| Number of comorbidities | 1.02 | 0.94 | 1.11 | 0.584 |
| Down syndrome diagnosis | 0.93 | 0.47 | 1.61 | 0.810 |
| Intellectual disability diagnosis | 0.50 | 0.34 | 0.70 | <0.0001 |

**Figure 1** Factors associated with interventions (non-invasive respiratory support, intubation and intensive care unit (ICU)) in hospitalised COVID-19 patients with and without an intellectual disability diagnosis. (A) Factors associated with access to non-invasive respiratory support . (B) Factors associated with the use of tracheal intubation. (C) Factors associated with admission to ICU. Bold labels on the forest plots indicate statistically significant associations. Per cent relative effects can be calculated using (risk ratio (RR)−1)×100 for RRs over 1 or (1−RR)×100 for RRs less than 1. For example, shortness of breath on admission was associated with a 73% ((1.73−1)×100) increase in risk of being admitted to the ICU while not requiring oxygen therapy of admission was associated with a 48% ((1−0.52)×100) decrease in risk of being admitted to the ICU while in hospital. We present log-transformed RRs in the plots.

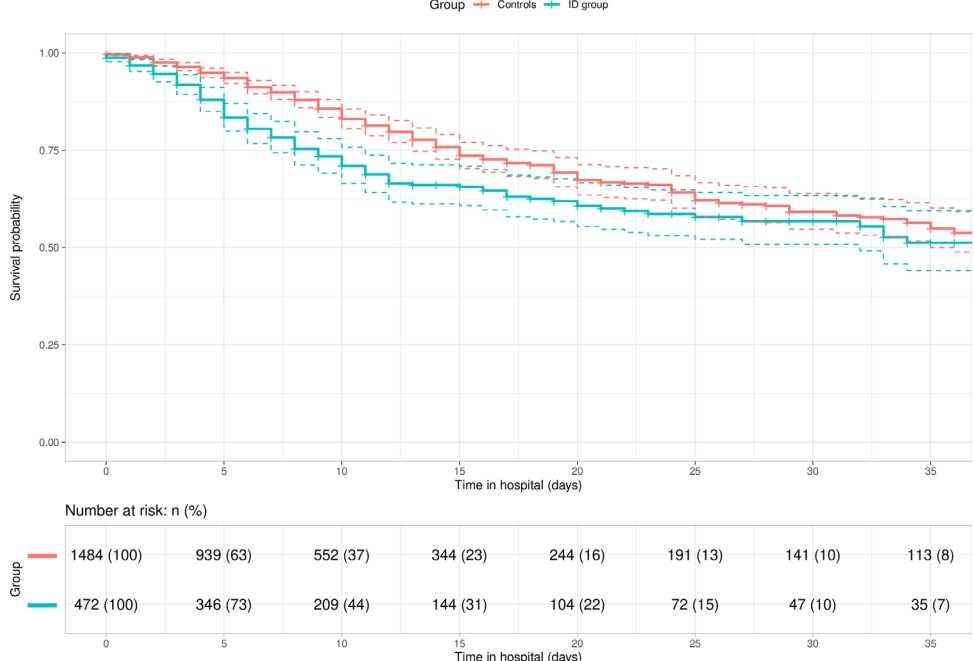

Number at risk: n (%)

| | | | | | | | |
|---|---|---|---|---|---|---|---|
| 1484 (100) | 939 (63) | 552 (37) | 344 (23) | 244 (16) | 191 (13) | 141 (10) | 113 (8) |
| 472 (100) | 346 (73) | 209 (44) | 144 (31) | 104 (22) | 72 (15) | 47 (10) | 35 (7) |

**Figure 2** Kaplan-Meier survival plot of hospitalised patients with COVID-19 with and without an intellectual disability diagnosis. ID, intellectual disabilities.

increased mortality rate in ID patients after admission to hospital, we found a different survival trajectory. ID patients died at a 1.44 times faster rate than the general population, even when age, comorbidities and severity of symptoms were considered. This suggests that aspects of

**Table 4** Factors associated with hospital length of stay in patients with COVID-19

| | Exp(β) | 95% CI | P value |
|---|---|---|---|
| *Age group* | | | |
| 20–29 years old | 1.23 | 1.10 to 1.37 | **0.0002** |
| 30–39 years old | 1.30 | 1.17 to 1.43 | **<0.0001** |
| 40–49 years old | 1.36 | 1.23 to 1.50 | **<0.0001** |
| 50–59 years old | 1.40 | 1.28 to 1.54 | **<0.0001** |
| 60–69 years old | 1.46 | 1.33 to 1.61 | **<0.0001** |
| 70–79 years old | 1.48 | 1.34 to 1.65 | **<0.0001** |
| 80+ years old | 1.69 | 1.49 to 1.92 | **<0.0001** |
| Male sex | 1.03 | 0.98 to 1.07 | 0.240 |
| Shortness of breath | 0.96 | 0.91 to 1.01 | 0.107 |
| Respiratory rate | 1.01 | 1.00 to 1.01 | **0.0003** |
| No oxygen therapy | 0.91 | 0.86 to 0.95 | **<0.0001** |
| Number of comorbidities | 1.05 | 1.04 to 1.07 | **<0.0001** |
| Down syndrome diagnosis | 1.08 | 0.95 to 1.22 | 0.229 |
| ID diagnosis | 1.15 | 1.09 to 1.22 | **<0.0001** |

Significant associations are highlighted in bold.
ID, intellectual disabilities.

their care and treatment may be contributing to increased mortality rather than comorbidities or complications.

People with ID who survived had a longer inpatient stay on average. Again, this does not appear to be secondary to increased complications or comorbidities. It is therefore possible that people with ID may be experiencing delays in their discharge and support to return to the community. Longer admissions can be associated with distress for the individual, exposure to risk of hospital acquired infections and institutionalisation. These findings highlight the different experiences of patients with ID after they were admitted to hospital for COVID-19 compared with the general population.

**Strengths and limitations**
The strengths of the study are the large sample size and the use of a well-matched control group which allows for comparisons in symptoms, treatment and outcomes to be captured. Data were taken from across the UK meaning it is reflective of experiences across the country rather than regionally specific issues. It used real-world data captured during an acute and evolving pandemic and gives insight into conditions faced by patients and health professionals at the time.

Some limitations are acknowledged. The study relied on data captured at the time of care. While this provides an accurate picture of acute clinical care, the nature of clinical records can lead to some degree of missing or incomplete data. In addition, the use of combined group categories (particularly the heterogenous group 'chronic neurological disorder') limited the ability to explore the potential impact of specific diseases, while the reason for specific clinical decisions may not be clear. Further

research is therefore needed to explore the details around clinical decision making for people with ID during pandemic conditions and the impact of care rationalisation on this population. It will also be important to understand the experiences of individuals with ID and role and experience of their caregivers, particularly with regards to decision making, advocacy and inclusion. As ISARIC4C CCP-UK is a UK population-based study and not specifically focused on people with ID, we were unable to consider the extent to which issues particularly relevant to people with ID such as availability of different modes of care, supported decision-making or the presence of family members or other close supportive persons to help with isolation and understanding of the pandemic may have affected our results. Further work is needed to examine how these factors may impact those admitted to hospital for COVID-19.

## CONCLUSION

These findings highlight an ongoing disparity in healthcare between people with ID and the general population which have been magnified by the COVID-19 pandemic, with implications for improving care and treatment during the ongoing crisis to ensure the levelling-up of services for the future. It is hard not to be concerned at the possibility of bias and discrimination affecting treatment decisions in such conditions, whether implicit or explicit. Barriers to care will need to be overcome and information should be disseminated in an accessible way to both caregivers and people with ID, particularly with regards to early symptoms and warning signs of a more severe presentation. In the community digital exclusion has been identified as a barrier to information for people with ID.[34] This may make it more difficult for people with ID to report early signs, receive up to date information about risks, or indeed even be part of track-and-trace systems. They may also be less able to self-monitor for early signs such as fevers. Moves towards the use of home oxygen saturation monitoring may be helpful in this population in identifying at risk people before they become acutely unwell and could allow for treatment to be initiated in a timely manner to reduce mortality.

Similarly, the results stress the need for people with ID admitted for COVID-19 (and other similar infections) to be prioritised for enhanced care and monitoring based on indicators of deterioration, without reliance on self-reporting. Earlier intervention may be required to avoid the more aggressive course of illness. Provisions and training should be in place in all hospitals regarding capacity and decision making, and trained staff should be available to assist in these matters. Echoing the recommendations of other researchers,[35] people with ID should be prioritised for COVID-19 vaccinations and boosters in the future. Care should be taken when making decisions about prioritisation of interventions to ensure they are not biased against people with long-term disabilities, but instead based on relevant prognostic indicators. Medical ethics panels which include professionals who are familiar with the care and needs of people with ID could assist with such decisions.

It is hoped that these results from the first wave of the pandemic highlight the ongoing health disparities faced by people with ID and will help raise awareness and mobilise healthcare services to improve practices and access for this population.

**Correction notice** This article has been corrected since it first published. Infographic supplementary file has been included.

**Acknowledgements** This report is independent research which used data provided by the MRC funded ISARIC 4C Consortium and which the Consortium collected under a research contract funded by the National Institute for Health Research. The views expressed in this publication are those of the authors and not necessarily those of the ISARIC 4C consortium.

**Contributors** AS conceived and designed the project with help from RAB. RAB and AS planned the data analysis. RAB conducted the data analysis. RAB, SEP and JS wrote the first draft with input from AS. All authors contributed to reviewing and revising the manuscript and agreed final approval of the version to be published.

**Funding** This study was supported by grants from the MRC (MR/S011277/1; MR/S005145/1; MR/R024901/1), the European Commission (H2020 SC1 Gene overdosage and comorbidities during the early lifetime in Down Syndrome GO-DS21-848077) and Alzheimer's Society (AS-CP-18-0020: fellowship to SEP). This study represents independent research part funded by the National Institute for Health Research (NIHR) Maudsley Biomedical Research Centre at South London and Maudsley NHS Foundation Trust and King's College London. The views expressed are those of the authors and not necessarily those of the NHS, the NIHR or the Department of Health and Social Care.

**Competing interests** None declared.

**Patient consent for publication** Not required.

**Ethics approval** Ethical approval was given by the South Central—Oxford C Research Ethics Committee in England (ref 13/SC/0149), the Scotland A Research Ethics Committee (ref 20/SS/0028) and the WHO Ethics Review Committee (RPC571 and RPC572, 25 April 2013).

**Provenance and peer review** Not commissioned; externally peer reviewed.

**Data availability statement** Data may be obtained from a third party and are not publicly available. The Independent Data and Material Access Committee welcomes applications for access to data and materials (https:// isaric4c.net).

**ORCID iD**
R Asaad Baksh http://orcid.org/0000-0001-6596-2145

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
