## [Reviewer comments · BMJ Open]

ARTICLE DETAILS

TITLE (PROVISIONAL)	Understanding inequalities in COVID-19 outcomes following hospital admission for people with Intellectual disability compared to the general population: A matched cohort study in the United Kingdom
AUTHORS	Baksh, R. Asaad; Pape, Sarah; Smith, James; Strydom, André

VERSION 1 – REVIEW

REVIEWER	De Cauwer, Harald University of Antwerp
REVIEW RETURNED	01-May-2021

GENERAL COMMENTS	This is a very interesting and promising manuscript but I would advise minor revision before considering publication. Some remarks: Page 3: Abstract: Maybe mention in results that controls express more cardiovascular risk, asthma, rheumatic disease and smoking. Page 5: Introduction: Those with Down syndrome may be at particular risk^{18,19} I would advise to mention other reports here as they contributed to the knowledge on this topic: De Cauwer H, Spaepen A. Are patients with Down syndrome vulnerable to life-threatening COVID-19? Acta Neurol Belg. 2020 May 22;1–3. doi: 10.1007/s13760-020-01373-8. Epub ahead of print. PMID: 32444942; PMCID: PMC7243430. This was the first article during the first wave to alarm medics that Down patients might be at higher risk. Clift AK, Coupland CAC, Keogh RH, Hemingway H, Hippisley-Cox J. COVID-19 Mortality Risk in Down Syndrome: Results From a Cohort Study of 8 Million Adults. Ann Intern Med. 2021 Apr;174(4):572-576. doi: 10.7326/M20-4986. Epub 2020 Oct 21. PMID: 33085509; PMCID: PMC7592804. Kantar A, Mazza A, Bonanomi E, Odoni M, Seminara M, Verde ID, Lovati C, Bolognini S, D'Antiga L. COVID-19 and children with Down syndrome: is there any real reason to worry? Two case reports with severe course. BMC Pediatr. 2020 Dec 18;20(1):561. doi: 10.1186/s12887-020-02471-5. PMID:
--

These manuscripts were the result of larger studies on Down and Covid and do suggest a more severe disease course in Down patients

Page 6: Participants and study design / page 7: study population and comorbidities

Please give the definition of Intellectual disability ID: I don't understand why people with multiple sclerosis, motor neuron disease, myasthenia should be considered as 'ID' , stroke and aphasia is not the same as ID.

Page 8/9: table1.

White: better Caucasian?

80+: in Belgium this was the predominant group in hospital during first wave: so it seems such a low figure in controls: method-related i.e. by matching controls to ID patients.

<20: can you find in the study why there are so many? Specific risk factors in this group? I 'm amazed that you found so many controls: also with higher percentage of risk factors than older patients??

>40: it has been shown that in Down patients >40 risk of more severe disease course is much higher. Maybe this can also be mentioned in the text .

Hüls A, Costa ACS, Dierssen M, Baksh RA, Bargagna S, Baumer NT, Brandão AC, Carfi A, Carmona-Iragui M, Chicoine BA, Ghosh S, Lakhanpaul M, Manso C, Mayer MA, Del Carmen Ortega M, de Asua DR, Rebillat AS, Russell LA, Sgandurra G, Valentini D, Sherman SL, Strydom A. An international survey on the impact of COVID-19 in individuals with Down syndrome. medRxiv [Preprint]. 2020; 5:2020.11.03.20225359.

Maybe mention in results that controls express more cardiovascular risk, asthma, rheumatic disease and smoking.(see also abstract)

Page 10

Subjective complaints less frequent in ID: due to history taking is less sufficient in ID patients? caregivers are not allowed in hospital or ER so hetero-anamnesis is not possible? because of history taking not possible in too dyspneic patients?

Page 10/11

More patients with ID and convulsions and confusion: new diagnosed seizures or already known with epilepsy but admitted to hospital because of more frequent seizures during COVID?

Interventions eg intubation less frequent in ID: do you have data on end of life decisions already made BEFORE hospitalization? So, not only different hospital policy in ID patients can explain the lower invasive interventions, but also restrictions already been implemented before by patients/caregivers.

Page 12

Faster mortality first 5 days: because of not applying invasive therapy?? Can your data rule this out: is this the group in which invasive techniques were not used??

Page 13:

Spending longer periods in hospitals: those who don't die the first 5 days: so outliers at left and right side of the curve, versus only outliers left side in controls??

Page 14:

Late presentation of patients with ID:

2 other possible explanations might be 1. Longer stay in facilities for people with ID before admitting to hospital because of disease severity 2. More rapid deterioration in patients with ID, eg Down syndrome.

Tolerability of interventions 'and thus insufficient adherence to treatment"

Do not resuscitate: do you have data on end of life decisions already made BEFORE hospitalization? So, not only different hospital policy in ID patients can explain the lower invasive interventions, but also restrictions already been implemented before by patients/caregivers. See also page 10/11

Faster mortality first 5 days: because of not applying invasive therapy?? Can your data rule this out: is this the group in which invasive techniques were not used?? (see also page 12)

Return to community: do you have data if patients with ID / controls come from home or from elderly homes, residential facilities for patients with ID?

Page 15/16 conclusion:

Please also mention these aspects:

There are some reports which state that people with ID have difficulties in adhering lockdown measures, eg Down patients still looking for contact with caregivers/family.

Patients with ID are more frequent living in residential facilities, so more prone to COVID transmission.

Patients with Down and other ID should be prioritized for vaccination!!!

Caregivers of patients with ID should be allowed in ER and in hospital to lower fear, unrest,..., and to get a better history taking and better en faster recognition of early warning signs.

Literature:

Courtenay K, Perera B. COVID-19 and People with Intellectual Disability: Impacts of a pandemic. Ir J Psychol Med. 2020;(2012).
Eshraghi AA, Li C, Alessandri M, Messinger DS, Eshraghi RS, Mittal R, et al. Correspondence - COVID-19: overcoming the

	challenges faced by individuals with autism and their families. The Lancet Psychiatry. 2020;7(June):481–3. Stevens M, De Cauwer H, Soontjens K, Spaepen A, Van Grieken S. Effect van de lockdown voor COVID-19 op bewoners van een residentiële voorziening. Signaal 2020; 113: 56-63 Dard R, Janel N, Vialard F. COVID-19 and Down’s syndrome: are we heading for a disaster? Eur J Hum Genet. 2020;3099(20):3–4. Xiang YT, Zhao YJ, Liu ZH, Li XH, Zhao N, Cheung T, et al. The COVID-19 outbreak and psychiatric hospitals in China: Managing challenges through mental health service reform. Int J Biol Sci. 2020;16(10):1741–4. Cammarata-Scalisi F, Tadich AC, Medina M, Callea M. Trisomy 21 and the coronavirus disease 2019 (COVID-19). Arch Argent Pediatr. 2020;118(4):230–1. Covid-19: What is happening with the vaccine rollout? Harald De Cauwer, Anneloes Rodiers, Ann Spaepen. BMJ 2021;372:n213 Del Carmen Ortega M, Borrel JM, de Jesús Bermejo T, González-Lamuño D, Manso C, de la Torre R, Mayer MA, de Asúa DR, Dierssen M; Spanish Trisomy 21 Research Society COVID-19 Taskforce. Lessons from individuals with Down syndrome during COVID-19. Lancet Neurol. 2020; 19(12):974-975 Is there a difference in policy and admittance of ID patients to ICU and getting intubated in hospital overwhelmed by COVID patients versus hospitals where the number of patients was manageable?
--	--

REVIEWER	Francis, Leslie The University of Utah, Law and Philosophy
REVIEW RETURNED	27-May-2021

GENERAL COMMENTS	This submission traces the hospital course of people with intellectual disabilities who are admitted with COVID. It compares these patients with matched controls and determines that the ID group had significantly lower rates of interventions, significantly longer hospital stays, and significantly higher rates of death. The interventions examined were non-invasive respiratory support, intubation, tracheostomy, ventilation, and ICU admission. Analysis was controlled for demographic variables, severity of illness on admission, and comorbidities related to COVID-19 outcomes. Analysis was also controlled for a diagnosis of Down. On admission, patients with ID had higher respiratory rates (suggesting higher disease severity), higher rates of altered consciousness, and lower reports of sensory change. These patients may have been referred only when their illness was more advanced possibly because of poor symptom recognition or communication difficulties. Questions/concerns: the AUs do not discuss whether reluctance to hospitalize or disability discrimination might have been part of the explanation for later referral to hospital. The AUs note that other studies have documented disparities in access to appropriate treatment for people with ID. Issues involved include decision-making capacity, ceilings imposed on care,
---

	inappropriate use of frailty scales, and discrimination. Longer inpatient stays may also be associated with lack of support in the community. While this study is useful data about the interventions and outcomes of people with ID hospitalized with COVID, it could be far more nuanced and informative. It does not consider the availability of modes of care particularly relevant to people with ID, such as supported decision-making or the presence of family members or other close supportive persons to help with isolation and understanding. In short, the AUs should acknowledge the extent to which other issues with the appropriate care people with ID received may have affected outcomes. It is for this reason that I checked the "no" boxes that I did--but I would have preferred a choice more like "not quite."
--	--

VERSION 1 – AUTHOR RESPONSE

Reviewer Reports:

Reviewer: 1

Dr. Harald De Cauwer, University of Antwerp

Comments to the Author:

This is a very interesting and promising manuscript but I would advise minor revision before considering publication.

Some remarks:

Reviewer Comment 1. Page 3: Abstract:

Maybe mention in results that controls express more cardiovascular risk, asthma, rheumatic disease and smoking.

Response: This has been added to the Abstract on Page 2:

‘Controls had higher rates of cardiovascular risk factors, asthma, rheumatologic disorder and smoking.’

Reviewer Comment 2. Page 5: Introduction:

Those with Down syndrome may be at particular risk^{18,19}

I would advise to mention other reports here as they contributed to the knowledge on this topic:

De Cauwer H, Spaepen A. Are patients with Down syndrome vulnerable to life-threatening COVID-19? Acta Neurol Belg. 2020 May 22:1–3. doi: 10.1007/s13760-020-01373-8. Epub ahead of print. PMID: 32444942; PMCID: PMC7243430. This was the first article during the first wave to alarm medics that Down patients might be at higher risk.

Clift AK, Coupland CAC, Keogh RH, Hemingway H, Hippisley-Cox J. COVID-19 Mortality Risk in Down Syndrome: Results From a Cohort Study of 8 Million Adults. Ann Intern Med. 2021 Apr;174(4):572-576. doi: 10.7326/M20-4986. Epub 2020 Oct 21. PMID: 33085509; PMCID: PMC7592804.

Kantar A, Mazza A, Bonanomi E, Odoni M, Seminara M, Verde ID, Lovati C, Bolognini S, D'Antiga L. COVID-19 and children with Down syndrome: is there any real reason to worry? Two case reports

with severe course. BMC Pediatr. 2020 Dec 18;20(1):561. doi: 10.1186/s12887-020-02471-5. PMID: These manuscripts were the result of larger studies on Down and Covid and do suggest a more severe disease course in Down patients

Response: We have incorporated these references to our introduction on Page 4:

'Those with Down syndrome may be at particular risk of a more severe disease course, ¹⁹⁻²¹ specifically those 40 years and older ²². Recent research has also suggested that people with Down syndrome also have an increased risk of COVID-19 hospitalisation and death ²³.'

Reviewer Comment 3. Page 6: Participants and study design / page 7: study population and comorbidities

Please give the definition of Intellectual disability ID: I don't understand why people with multiple sclerosis, motor neuron disease, myasthenia should be considered as 'ID', stroke and aphasia is not the same as ID.

Response: The neurological disorders diagnosis which was used by ISARIC4C CCP-UK was a broad category including cerebral palsy, multiple sclerosis, motor neurone disease, muscular dystrophy, myasthenia gravis, Parkinson's disease, stroke and severe learning difficulty. We did not include this in our definition of Intellectual disability. However, we have added a clearer definition of ID to our introduction on Page 4:

'Intellectually disability (ID) is a condition characterized by varying degrees of impairments in cognition, language, motor and social abilities depending on the severity of ID ^{1'}

Reviewer Comment 4. Page 8/9: table1.

White: better Caucasian?

80+: in Belgium this was the predominant group in hospital during first wave: so it seems such a low figure in controls: method-related i.e. by matching controls to ID patients.

<20: can you find in the study why there are so many? Specific risk factors in this group? I'm amazed that you found so many controls: also with higher percentage of risk factors than older patients??

>40: it has been shown that in Down patients >40 risk of more severe disease course is much higher. Maybe this can also be mentioned in the text .

Hüls A, Costa ACS, Dierssen M, Baksh RA, Bargagna S, Baumer NT, Brandão AC, Carfi A, Carmona-Iragui M, Chicoine BA, Ghosh S, Lakhanpaul M, Manso C, Mayer MA, Del Carmen Ortega M, de Asua DR, Rebillat AS, Russell LA, Sgandurra G, Valentini D, Sherman SL, Strydom A. An international survey on the impact of COVID-19 in individuals with Down syndrome. medRxiv [Preprint]. 2020; 5:2020.11.03.20225359.

Maybe mention in results that controls express more cardiovascular risk, asthma, rheumatic disease and smoking.(see also abstract)

Response: The ethnicity term White was used on the data collection forms in keeping with the UK ethnicity classification when patients were admitted to hospital, consequently we have chosen to use White to describe this group.

80+: The smaller 80+ age control group size is an artefact of our age, gender and ethnicity matching to our ID group.

<20: We think this is due to selection bias based on the data that we have analysed. Our analysis was conducted on hospital admission data and therefore only those <20 who were extremely unwell from COVID-19 would have been admitted to hospital and included in our study.

We are unsure what Dr De Cauwer means by 'also with higher percentage of risk factors than older patients' since we have not provided comorbidities on page 8/9 by age group, but rather group (controls vs ID patients).

We had included the Hul et al (2021) paper in our introduction but we have now highlighted the >40 years old findings as suggested on Page 4:

'Those with Down syndrome may be at particular risk of a more severe disease course, ¹⁹⁻²¹ specifically those 40 years and older ²²'

Reviewer Comment 5. Page 10

Subjective complaints less frequent in ID: due to history taking is less sufficient in ID patients? caregivers are not allowed in hospital or ER so hetero-anamnesis is not possible? because of history taking not possible in too dyspneic patients?

Response: We agree that these are all possible explanations, and have covered most of these possibilities in the discussion already.

Reviewer Comment 6. Page 10/11

More patients with ID and convulsions and confusion: new diagnosed seizures or already known with epilepsy but admitted to hospital because of more frequent seizures during COVID?

Response: Specific data on seizures and epilepsy as a comorbidity prior to admission to hospital for COVID-19 was not collected. We only had access to data on seizures as a sign and symptom of COVID-19 at point of admission, therefore it would be difficult to say whether the presentation of convulsions at admission is an exacerbation of underlying vulnerability to seizures based on the data we have.

Reviewer Comment 7. Interventions eg intubation less frequent in ID: do you have data on end of life decisions already made BEFORE hospitalization? So, not only different hospital policy in ID patients can explain the lower invasive interventions, but also restrictions already been implemented before by patients/caregivers.

Response: We do not have access to data on end of life decision made before admission to hospital. However, we thank Dr De Cauwer for this important observation.

Reviewer Comment 8. Page 12

Faster mortality first 5 days: because of not applying invasive therapy?? Can your data rule this out: is this the group in which invasive techniques were not used??

Response: We explored this hypothesis through an (unreported) mediation analysis but as there may also be other currently unknown factors which may explain the faster mortality rates, we did not think this analysis contributed much to the paper.

Reviewer Comment 9. Page 13:

Spending longer periods in hospitals: those who don't die the first 5 days: so outliers at left and right side of the curve, versus only outliers left side in controls??

Response: The data we presented is correct for the participants included in analysis. Control sample consists of matched individuals so not necessarily representative of the general population.

Reviewer Comment 10. Page 14:

Late presentation of patients with ID:

2 other possible explanations might be 1. Longer stay in facilities for people with ID before admitting to hospital because of disease severity 2. More rapid deterioration in patients with ID, eg Down syndrome.

Tolerability of interventions 'and thus insufficient adherence to treatment'

Do not resuscitate: do you have data on end of life decisions already made BEFORE hospitalization? So, not only different hospital policy in ID patients can explain the lower invasive interventions, but also restrictions already been implemented before by patients/caregivers. See also page 10/11

Faster mortality first 5 days: because of not applying invasive therapy?? Can your data rule this out: is this the group in which invasive techniques were not used?? (see also page 12)

Return to community: do you have data if patients with ID / controls come from home or from elderly homes, residential facilities for patients with ID?

Response: We agree that late presentation may be both due to patients with ID having delayed referral to hospital, and also, potentially more rapid deterioration. We do not have sufficient detail on patients' accommodation to comment on whether they were resident in ID facilities or not.

Reviewer Comment 11. Page 15/16 conclusion:

Please also mention these aspects:

There are some reports which state that people with ID have difficulties in adhering lockdown measures, eg Down patients still looking for contact with caregivers/family.

Patients with ID are more frequent living in residential facilities, so more prone to COVID transmission.

Patients with Down and other ID should be prioritized for vaccination!!!

Caregivers of patients with ID should be allowed in ER and in hospital to lower fear, unrest,..., and to get a better history taking and better en faster recognition of early warning signs.

Literature:

Courtenay K, Perera B. COVID-19 and People with Intellectual Disability: Impacts of a pandemic. *Ir J Psychol Med.* 2020;(2012).

Eshraghi AA, Li C, Alessandri M, Messinger DS, Eshraghi RS, Mittal R, et al. Correspondence - COVID-19: overcoming the challenges faced by individuals with autism and their families. *The Lancet Psychiatry.* 2020;7(June):481–3.

Stevens M, De Cauwer H, Soontjens K, Spaepen A, Van Grieken S. Effect van de lockdown voor COVID-19 op bewoners van een residentiële voorziening. *Signaal* 2020; 113: 56-63

Dard R, Janel N, Vialard F. COVID-19 and Down's syndrome: are we heading for a disaster? *Eur J Hum Genet.* 2020;3099(20):3–4.

Xiang YT, Zhao YJ, Liu ZH, Li XH, Zhao N, Cheung T, et al. The COVID-19 outbreak and psychiatric hospitals in China: Managing challenges through mental health service reform. *Int J Biol Sci.* 2020;16(10):1741–4.

Cammarata-Scalisi F, Tadich AC, Medina M, Callea M. Trisomy 21 and the coronavirus disease 2019 (COVID-19). *Arch Argent Pediatr.* 2020;118(4):230–1.

Covid-19: What is happening with the vaccine rollout? Harald De Cauwer, Anneloes Rodiers, Ann Spaepen. *BMJ* 2021;372:n213

Del Carmen Ortega M, Borrel JM, de Jesús Bermejo T, González-Lamuño D, Manso C, de la Torre R, Mayer MA, de Asúa DR, Dierssen M; Spanish Trisomy 21 Research Society COVID-19 Taskforce. Lessons from individuals with Down syndrome during COVID-19. *Lancet Neurol.* 2020; 19(12):974-975

Response: We thank the reviewer for these suggestions, but due to word limitations, we cannot include these, particularly since it relates to susceptibility and exposure to infection, whereas this paper is focussed on treatment in hospital. We agree however that patients with ID should be a priority group for vaccination and we have added the following to our discussion on Page 15 in regards to prioritisation of vaccination for people with ID:

'Echoing the recommendations of other researchers³⁴, people with ID should be prioritised for COVID-19 vaccinations and boosters in the future.'

Reviewer Comment 12. Is there a difference in policy and admittance of ID patients to ICU and getting intubated in hospital overwhelmed by COVID patients versus hospitals where the number of patients was manageable?

Response: We do not have data on whether hospitals included in the study were overwhelmed by COVID-19 patients and whether this may have influenced decisions on admittance of ID patients to ICU and getting intubated. However, this is an important question which is worth exploring in future studies.

Reviewer: 2
Dr. Leslie Francis, The University of Utah

Comments to the Author:

This submission traces the hospital course of people with intellectual disabilities who are admitted with COVID. It compares these patients with matched controls and determines that the ID group had significantly lower rates of interventions, significantly longer hospital stays, and significantly higher rates of death. The interventions examined were non-invasive respiratory support, intubation, tracheostomy, ventilation, and ICU admission. Analysis was controlled for demographic variables, severity of illness on admission, and comorbidities related to COVID-19 outcomes. Analysis was also controlled for a diagnosis of Down. On admission, patients with ID had higher respiratory rates (suggesting higher disease severity), higher rates of altered consciousness, and lower reports of sensory change. These patients may have been referred only when their illness was more advanced possibly because of poor symptom recognition or communication difficulties.

Reviewer Comment 1. Questions/concerns: the AUs do not discuss whether reluctance to hospitalize or disability discrimination might have been part of the explanation for later referral to hospital. The AUs note that other studies have documented disparities in access to appropriate treatment for people with ID. Issues involved include decision-making capacity, ceilings imposed on care, inappropriate use of frailty scales, and discrimination. may also be associated with lack of support in the community.

Response: We have included this suggestion in our discussion on Page 13:

‘Other issues which may have contributed to later referral to hospital include a reluctance from family members to hospitalise their relative or disability discrimination resulting in people with ID not being able to access medical services.’

Reviewer Comment 2. While this study is useful data about the interventions and outcomes of people with ID hospitalized with COVID, it could be far more nuanced and informative. It does not consider the availability of modes of care particularly relevant to people with ID, such as supported decision-making or the presence of family members or other close supportive persons to help with isolation and understanding. In short, the AUs should acknowledge the extent to which other issues with the appropriate care people with ID received may have affected outcomes. It is for this reason that I checked the "no" boxes that I did--but I would have preferred a choice more like "not quite."

Response: We thank Dr Francis for this valuable consideration, however we were unable to examine these important factors in the present study because such detail was not included in the WHO ISARIC-4C survey. Nevertheless, we have added Dr Francis’s point to our limitation section on Page 15:

‘Moreover, as ISARIC4C CCP-UK is a UK population-based study and not specifically focused on people with ID, we were unable to consider the extent to which issues particularly relevant to people with ID such as availability of different modes of care, supported decision-making or the presence of family members or other close supportive persons to help with isolation and understanding of the pandemic may have affected our results. Further work is needed to examine how these factors may impact those admitted to hospital for COVID-19.’

VERSION 2 – REVIEW

REVIEWER	De Cauwer, Harald University of Antwerp
REVIEW RETURNED	11-Jul-2021

GENERAL COMMENTS	nice work, very important for future allocation of care discussion the recommendations of the first review have been cared for in the manuscript. So the manuscript can be published without any further corrections. Fantastic job done!
---

REVIEWER	Francis, Leslie The University of Utah, Law and Philosophy
REVIEW RETURNED	31-Jul-2021

GENERAL COMMENTS	This is a well done and critically important study. It should definitely be published. My one comment is that I would like to see the authors reflect more fully on whether and how the ability to tolerate treatment might have contributed to differences in hospital course. I don't know whether the data admit of this—but it could be useful to consider, for example, whether any support persons were able to be with the person with ID in hospital, to help them deal with poorly-understood and disturbing invasive care. The authors appropriately acknowledge the study limits and hope that it will serve to spur future research. That research is sorely needed and hopefully this study will serve to encourage it.
--